# RECORD: a Question Answering tool for COVID-19

**Matteo Muffo, Aldo Cocco, Mattia Messina, Enrico Bertino**
Indigo.ai
Via Torino 61, Milan, Italy
{matteo, aldo, mattia, e}@indigo.ai

## Abstract

In this work we present RECORD (Research Engine for COVID Open Research Dataset), a tool to provide answers to questions related to COVID-19. The engine that we propose adopts a two-step pipeline to extract answers from the CORD-19 dataset, the most complete collection of scientific articles about COVID-19 disease and SARS-CoV-2. RECORD leverages a large pre-trained language model to select, from a given input question, the most semantically-related papers, then a Question Answering model is used to provide the best answer for each selected paper. In order to evaluate the performances of our tool, we tested it on two sets of questions (one proposed by our evaluation team plus one proposed by the health facility *Centro Medico Sant'Agostino*), adopting a scoring mechanism to assess the quality of the outcomes. In this context, RECORD has shown remarkable results in terms of consistency and precision of the extracted answers.

## 1 Introduction

As the spreading of the pandemic urged entire nations to the state of emergency, the medical and scientific community found themselves with an enormous pressure for the need of solutions to a problem still largely unknown to the public. Following the attention that this matter gained, researchers from all over the world started to publish new studies that contributed to expand even more the existing literature. As a result, the community itself couldn't keep up with the same pace the literature was growing. Fruition became a crucial issue.

We responded to this problem building RECORD, a tool to provide answers to questions related to COVID-19.

This engine takes one or more questions as input and is able to go through the entire dataset to successfully provide answers drawn from the paper's

text. We decided to use the COVID-19 Open Research Dataset (CORD-19) (Wang et al., 2020) as source of information for our tool. The CORD-19 dataset consists of a collection of research articles, arranged by The White House and the Allen Institute of AI, jointly with a coalition of leading research groups. It includes, at the time of writing, over 75,000 articles with full text, containing the most recent studies conducted on the COVID-19 disease as well as a large portion concerning previous related coronaviruses, like SARS and MERS.

Our tool adopts a pipeline composed of two main sections: firstly, it filters out a subset of most semantically-related papers to an input query. We designed this step using a Sentence-BERT model (Reimers and Gurevych, 2019), an algorithm based on a BERT architecture to generate a representation of a piece of text. It calculates similarities between papers and the input query and can then filter the most similar ones. Lastly, it uses a BERT-based Question Answering model which receives as input a paper and a query, and is able to provide answers extrapolated from the body of the article. This last section is performed until the whole subset of selected papers is processed.

## 2 The pipeline

In this section we will extensively illustrate the pipeline used by RECORD to provide answers [1].

### 2.1 Preprocessing

Since the purpose of RECORD is to answer questions specifically related to COVID-19, we ignore articles in the CORD-19 dataset about similar viruses like SARS or MERS, focusing only on those about COVID-19 and SARS-CoV-2. In order to do that, we filtered data according to the

---

[1]Code available at: https://www.kaggle.com/matteomuffo/a-fine-grained-covid-19-question-answering-engine

publishing year (only papers published since 2019 are considered) and a list of keywords identifying COVID-19 virus.

The main part of the preprocessing step consists of splitting the body texts into chunks: the Sentence-BERT network has a maximum input sequence length of 128 tokens, so it is necessary to separate the text into paragraphs, without losing the semantic relationship inside them. For this reason, we chose a fixed number of tokens $k$ (slightly smaller than 128) and we constructed the first paragraph taking the first $k$ tokens (after text tokenization) and then extended (or cut) the paragraph in correspondence of the closest token that represents a point (it can be before or after the $k$-th token). In this way, even if it appears after the $k$-th token, we are sure to have paragraphs with no more than 128 tokens, because we chose $k$ sufficiently smaller than 128.

We iterate the process until the end of text, obtaining the chunks representation for each paper $B_i$:

$$B_i = (C_{i1}, C_{i2}, \ldots, C_{in_i})$$

with $n_i$ being the number of chunks in the paper $B_i$.

This approach leads to a computationally expensive embedding step (due to the high number of embedding vectors to compute), but allows to perform a fine-grained analysis of all the available data, searching at chunk-level the papers which are most semantically-related to the query.

## 2.2 Embedding step

In order to generate the most appropriate answers to a given query, the first step is to extract a set of most semantically-correlated papers, where it is more likely to find an answer. All the chunks of the body texts in the dataset are embedded using the *Sentence-BERT* model (Reimers and Gurevych, 2019).

When a query is given as input, it is embedded using the same Sentence-BERT network; then, for each paper $i$, we compute the cosine similarities between the sentence embedding relative to the query $\mathbf{v}_Q$ and the sentence embeddings associated to the chunks of the paper, $\mathbf{v}_{C_{ij}}$:

$$\cos(\mathbf{v}_Q, \mathbf{v}_{C_{ij}}) = \frac{\mathbf{v}_Q \cdot \mathbf{v}_{C_{ij}}}{||\mathbf{v}_Q|| \, ||\mathbf{v}_{C_{ij}}||}.$$

After this computation we store the maximum value of cosine similarity achieved by each paper: $M_i = \max_j(\cos(\mathbf{v}_Q \cdot \mathbf{v}_{C_{ij}}))$. In this way,

we use the information related to the entire body text. The distance between the input query and each paper is equal to the distance between the query and the closest chunk. It is simple now to extract the most semantically-related papers with respect to the query by selecting the $N$ biggest values within the set $\{M_1, M_2, \ldots, M_K\}$, with $K$ being the number of papers in the dataset.

## 2.3 Question Answering step

Once the subset of most relevant papers is selected, the goal is to extract an answer for each paper. A Question Answering model (QA) is used, based on a $BERT_{LARGE}$ architecture pre-trained on a general-domain corpus and fine-tuned on the *Stanford Question Answering Dataset* (SQuAD) 1.1 dataset (Rajpurkar et al., 2016). The model takes in input the query and the context in the usual BERT format:

$$[CLS] \; \underbrace{L_1 \; \ldots \; L_{T_1}}_{\text{query}} \; [SEP] \; \underbrace{X_1 \; \ldots \; X_{T_2}}_{\text{context}} \; [SEP]$$

where $(L_1, \ldots, L_{T_1})$ are the $T_1$ tokens of the input query and $(X_1, \ldots, X_{T_2})$ are the $T_2$ tokens of the context. As output, the model generates the span of text that better represents the answer to the question. It is assumed that the answer is always contained in the context.

For the QA model, we use again the chunks obtained in Section 2.2. For a given paper, the QA engine receives as input the initial query and the paper in separated chunks: the model processes each chunk and provides the best answer together with a score. Once doing it for all the chunks of the paper, a list of answers and associated scores is obtained. At the end, the model identifies the best answer for the considered paper as the one obtaining the highest score among all the processed chunks. The same procedure is repeated for all the papers selected in the previous step. The score used to discriminate the best answer in a paper consists of the sum of the logit probabilities of the start and end tokens identified by the QA model.

It is important to underline that the best answer is generated by processing all the chunks of the paper, not only the one achieving maximum cosine similarity in the embedding step.

To highlight the reliability of the answer, we extract additional information related to the paper. In particular, RECORD provides:

- *Title*

- *Authors*

- *Publishing journal*

- *Scimago Journal Score*: it is a measure of journal's impact, influence or prestige. It expresses the average number of weighted citations received in the selected year by the documents published in the journal in the three previous years (https://www.scimagoir.com)

- *Paper citations*: influential citations count of each paper, provided by *Google Scholar*

- *Level of Evidence*: level of the hierarchy of evidence of the experiment described in the paper. This value is obtained via a keywords-based approach.

## 3  Review of alternatives

In this section we want to give an outline of the models that we adopted in our pipeline, together with a review of the alternatives we tested.

In both the steps described in Section 2, RECORD relies on a BERT network (Devlin et al., 2018) to accomplish the task. For the information retrieval step, we filter the papers most related to the query adopting an embedding-based approach that leverages the semantic information encoded in universal sentence representations. Among the alternatives, our choice fell on Sentence-BERT (SBERT) (Reimers and Gurevych, 2019), i.e. a BERT network fine-tuned on two natural language inference datasets with the aim to obtain semantically meaningful sentence representations. SBERT represents the sentence embedding model that achieves the state of the art in the *Semantic Textual Similarity* shared task series (Agirre et al., 2012, 2013, 2014, 2015, 2016; Cer et al., 2017). As an alternative to SBERT, we tested the BM25 Okapi algorithm (Robertson and Zaragoza, 2009). The main differences between the two approaches can be resumed in two points:

- while SBERT encodes information in sentence embeddings, BM25 encodes sentences in Bag of Words vectors, involving all the problems related to curse of dimensionality and feature redundancy associated to this type of representation.

- SBERT relies on a big neural network with millions of parameters, while BM25 is a non-parametric algorithm. As a result, the computational effort required by SBERT is much higher compared to BM25.

As expected, we empirically noticed that the sentence embedding approach (SBERT) performs better in this context.

Once our choice fell on SBERT to accomplish the information retrieval task, we explored different variants of architecture and pre-trainings. In particular, the alternatives that we tested are:

- *CovidBERT*, pre-trained on Allen AI's CORD-19 dataset (Wang et al., 2020), and fine-tuned on SNLI (Bowman et al., 2015) and MultiNLI datasets (Williams et al., 2018)

- *BioBERT* (Lee et al., 2019), pre-trained on large scale biomedical corpora and fine-tuned on SNLI and MultiNLI datasets

- *SciBERT* (Beltagy et al., 2019), pre-trained on a large multi-domain corpus of scientific publications and fine-tuned on SNLI and MultiNLI datasets

- *DistilBERT* (Sanh et al., 2019), a lightweight BERT network pre-trained on a general domain corpus and fine-tuned on SNLI, MultiNLI and STS-B datasets.

Evaluating empirically the results provided by each of the models listed, we concluded that DistilBERT was the best alternative. Our opinion with respect to this result is that although models trained on domain-specific corpora can better represent specific terms, the amount of training data is not enough to achieve comparable performances with respect to DistilBERT, which is pre-trained on a bigger general-domain corpus.

For what concerns the Question Answering task, we tested several BERT networks fine-tuned on the SQuAD datasets, varying pre-training domain or architecture (similarly for what we did reviewing the alternatives of SBERT). The proposals that we analysed are:

- *CovidBERT* fine-tuned on the SQuAD 2.0 dataset (Rajpurkar et al., 2018)

- *BioBERT* fine-tuned on the SQuAD 2.0 dataset

- *SciBERT* fine-tuned on the SQuAD 2.0 dataset

- $BERT_{LARGE}$, pre-trained on a general-domain corpus and fine-tuned on the SQuAD 1.1 dataset (Rajpurkar et al., 2016).

Also in this case, the model trained on a general-domain corpus resulted to be the best alternative, so we adopted $BERT_{LARGE}$ in our pipeline for the Question Answering step.

We underline that all the comparisons described in this section are conducted empirically, without a quantitative metric that can evaluate the quality of the tested models in the task. We opted for this type of evaluation for lack of time and resources. A quantitative comparison can be a future improvement.

## 4 Results

In this section we will describe the scoring mechanism that we proposed to evaluate the performances of RECORD and the results obtained on a testset of questions.

### 4.1 Score description

We provide a score metric to evaluate the quality of the generated answers in terms of precision and consistency with the input question. We asked humans to annotate a score for each provided answer, going from 0 to 3.

- *Score 0: Wrong Topic*
  The answer topic is different from the question topic.

- *Score 1: Wrong Answer*
  The topic is correct, but the text does not answer the question.

- *Score 2: Generic Answer*
  The topic is correct, but the answer is generic and not precise.

- *Score 3: Specific Answer*
  The answer is consistent and precise.

### 4.2 Tasks description and testset

We evaluated RECORD on a list of key questions, divided by tasks: some of them are more specific, like potential COVID-19 risks factors or what is known about transmission and incubation of the virus. Other tasks are more generic: we want to know what has been published concerning ethical and social science considerations, or what we know about the effectiveness of non-pharmaceutical interventions. In this way, we are testing how the tool behaves with both narrowed questions and more complex topics.

The testset has a total size of 112 questions. Some questions are drawn from the *NASEM's SCIED* (National Academies of Sciences, Engineering, and Medicine's Standing Committee on Emerging Infectious Diseases and 21st Century Health Threats [2]) research topics and the *World Health Organization's R&D Blueprint for COVID-19* [3]. Other questions have been created by our evaluation team to simulate the search process of a specialized user as best as possible. Finally, a subset of questions has been produced and scored by specialists of *Centro Medico Sant'Agostino* [4], a partner facility in the healthcare industry. In Appendix A we report all the questions submitted to RECORD.

### 4.3 Results and discussion

Figure 1 reports bar plots aggregating the scores assigned to the answers provided by RECORD. In particular, to better expose the performances of our tool, we propose three charts: the first one collects the scores obtained by the first answer for each question, the second collects the maxima among the first three answers provided for each question, while the third reports the maxima among all $N$ answers provided for each question. We decided to take $N$=5 answers for each question. The principle behind the choice of reporting the maxima consists in the fact that we want to highlight RECORD's performances in providing at least one good answer. Moreover, we recall that the answers of a question are provided in decreasing order of cosine similarity with respect to the query (in the embedding step of the engine), and for this reason we expect that the human-assigned scores of the answers accordingly decrease (to be verified in future works).

Figure 1a, in which we only consider the first answer, shows that RECORD is able to provide consistent and precise answers $32, 3\%$ of times (score 3), while there is a lower percentage ($21, 9\%$) of answers obtaining score 0. Figure 1b shows that, con-

---

[2]https://www.nationalacademies.org/our-work/standing-committee-on-emerging-infectious-diseases-and-21st-century-health-threats

[3]https://www.who.int/

[4]https://www.cmsantagostino.it/it

sidering the maximum among the first 3 answers for each question, the percentage of answers obtaining score 3 grows significantly, reaching 51%. Lastly, figure 1c evidences a further increase in the quality of the answers considering the maximum score among all the 5 answers provided for each question. In particular, $62, 5\%$ of the maximums have score 3 and just in the $5, 21\%$ of the cases all answers relative to a question are off-topic (maximum score 0).

In order to better understand how the performances of RECORD vary with respect to the topic of the query, in figure 2 we report bar plots collecting the maxima among all answers for each question, divided by task. One can notice that the task 9, addressing population studies, obtains maximum score 1 for $50\%$ of the answers and represents the worst performance among all tasks analyzed. A possible explanation of this result can be identified in a low coverage of the topic in the CORD dataset. For all other tasks, RECORD obtains a score of 2 or 3 for most of the answers.

About the scores assigned by *Centro Medico Sant'Agostino* (figure 3), we notice that also in this case most part of the maxima ($50\%$) achieves score 3, but there is a substantial part of answers which obtains a maximum score of 0 or 1 (37.5%). This can be due to very specific questions not covered by the CORD Dataset (such as *"What is the false positive and false negative rate in Diasorin serological rapid test?"* which is referring to a particular diagnosis instrument). A possible improvement to this work could be the introduction of a disclaimer for the user if the topic of the question is not included in the literature.

To conclude the discussion about our tool, we believe that a substantial improvement to RECORD's performances is brought by the analysis of the whole body of the articles, performed both in the embedding and Question Answering steps. Although a higher computational effort, the chunking operation described in Section 2.1 allows RECORD to operate a detailed inspection of the entirety of the information available.

## 5 Conclusions

In this work we presented RECORD, a tool to provide answers to COVID-related questions extracting information from the papers contained in the CORD-19 dataset. We showed that the two-step pipeline together with the fine-grained inspection, performed thanks to the chunking operation, led our tool to reach remarkable performances. Relying on a scoring mechanism to evaluate the quality of the answers, we showed that RECORD is able to provide at least one consistent and precise answer $61, 3\%$ of the time. The performances are similar even when the scoring is performed by external and unbiased medical researchers from *Centro Medico Sant'Agostino*. This is a very good achievement given the complex nature of the topic and the possible incompleteness of the dataset.

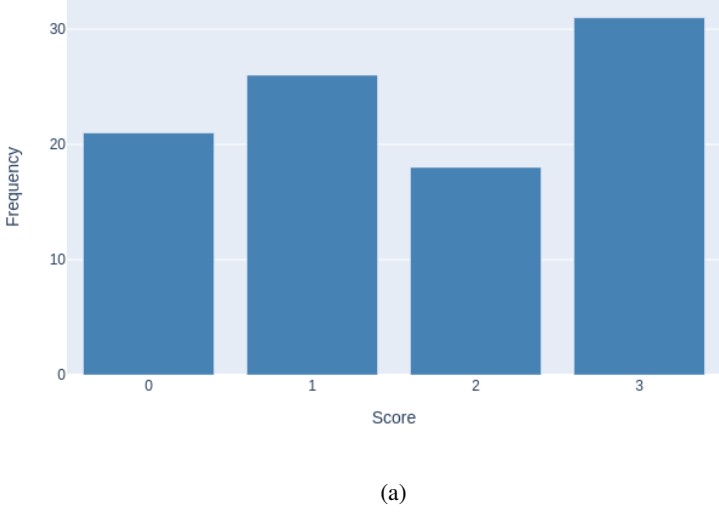

(a)

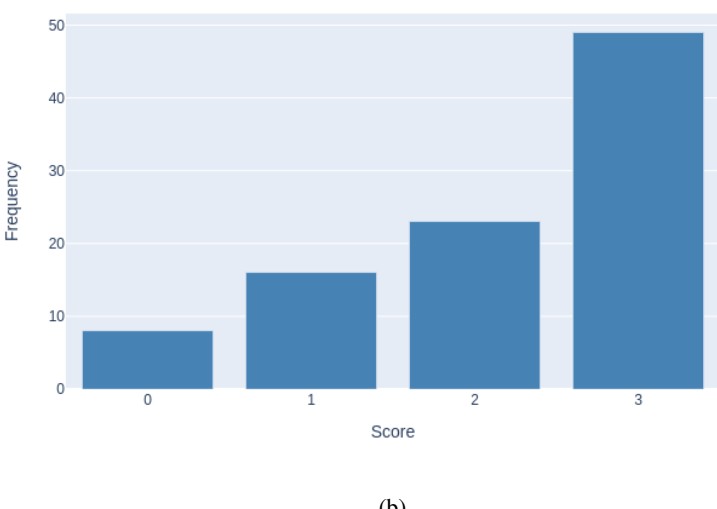

(b)

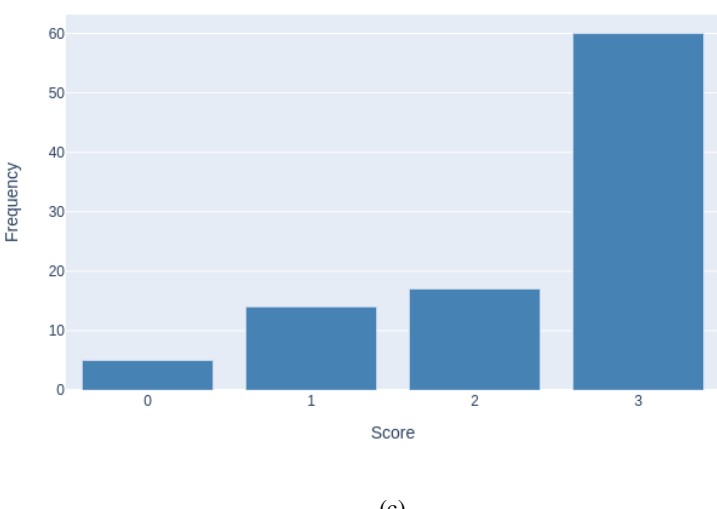

(c)

Figure 1: Bar plot representing the maximum scores: Figure 1a represents the maximum scores obtained by the first answer of each query. Figure 1b represents the maximum scores computed among the first three answers of each query. Figure 1c represents the maximum scores computed over all answers of each query.

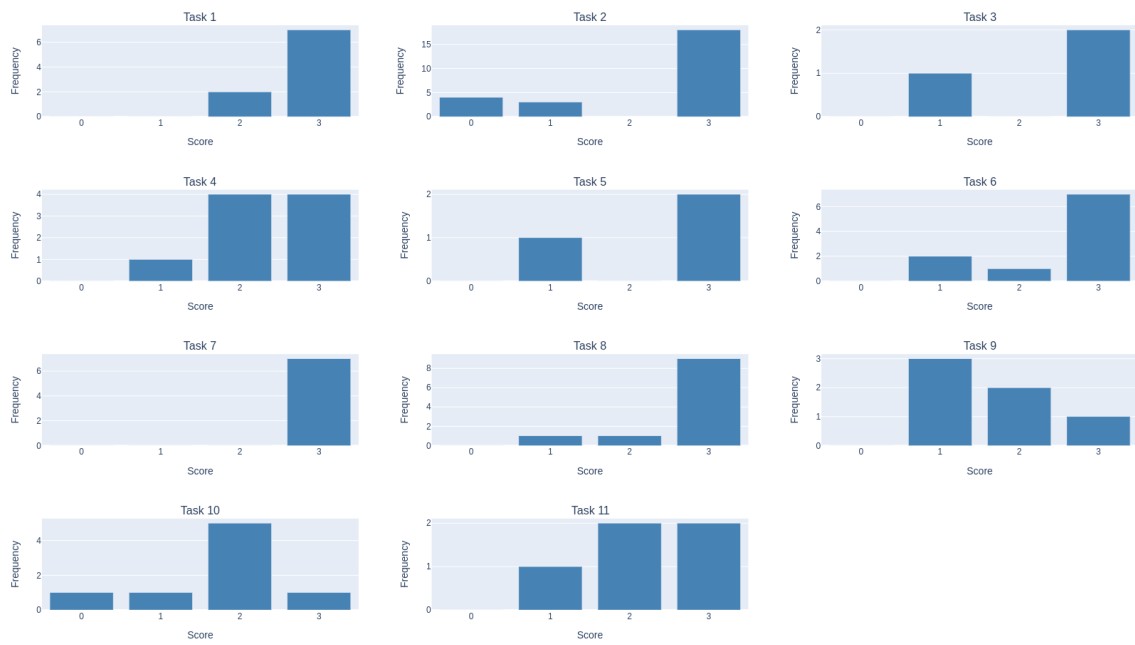

Figure 2: Bar plots of the maximum scores computed over all answers of each query, divided by task

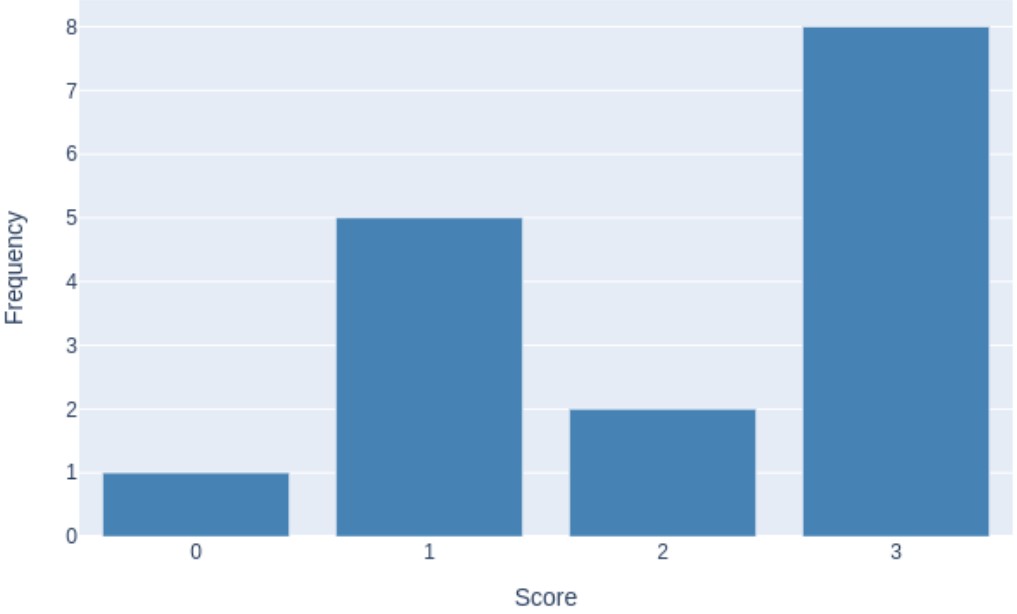

Figure 3: Bar plot of the maximum scores computed over all answers of the queries formulated and scored by *Centro Medico Sant'Agostino*

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

# A Appendices

## A.1 Questions

In the following tables we report the entire list of questions submitted to RECORD.

| Task | Question |
|---|---|
| Task 1:
Transmission, incubation
and environmental stability | Are movement control strategies effective?
Are there diagnostics to improve clinical processes?
Does the environment affect transmission?
How long does the virus persist on surfaces?
How long individuals are contagious?
Is personal protective equipment effective?
What is known about immunity?
What is the natural history of the virus?
What is the range of the incubation period in humans? |
| Task 2:
COVID-19 risk factors | Are co-infections risk factors?
Are male gender individuals more at risk for COVID-19?
Are pulmunary diseases risk factors?
Are there any public health mitigation measures considered effective?
Do we consider chronic kidney disease a risk factor for COVID-19?
Do we consider chronic respiratory diseases risk factors for COVID-19?
Do we consider drinking a potential risk factor for COVID-19?
Do we consider respiratory system diseases a risk factor for COVID-19?
How does chronic liver disease increases the risk for COVID-19?
How does obesity increases the risk for COVID-19?
How does overweight increases the risk for COVID-19?
Is cancer a risk factor for COVID-19?
Is cardio-cerebrovascular disease a risk factor for COVID-19?
Is cerebrovascular disease a risk factor for COVID-19?
Is individual's age considered a potential risk factor?
Is smoking a risk factor?
What do we know about risk factors related to COPD?
What do we know about risk factors related to Diabetes?
What do we know about risk factors related to heart diseases?
What do we know about risk factors related to hypertension?
What is the basic reproductive number?
What is the serial interval?
What is the severity of the disease?
Which are high-risk patient groups?
Which are the environmental risk factors? |

Table 1: Questions submitted to RECORD for the tasks 1 and 2.

| Task | Question |
|---|---|
| Task 3:
Vaccines, therapeutics,
interventions and clinical
studies | Are there any drugs proven to be effective in treating COVID-19 patients?

What is the best method to combat the hypercoagulable state seen in COVID-19?

What is the efficacy of novel therapeutics being tested currently? |
| Task 4: Diagnostics and
surveillance | Are there diagnosis techniques based on antibodies?
Are there diagnosis techniques based on nucleic-acid tech?
Are there new advances in diagnosing SARS-COV-2?
Are there point-of-care tests being developed?
Are there rapid bed-side tests?
How does viral load relate to disease presentations?
How does viral load relate to likelihood of a positive diagnostic test?
Is there any policy or protocol for screening and testing?
What do we know about diagnostics and coronavirus? |
| Task 5:
How geography affects virality | Are there geographic variations in the mortality rate of COVID-19?
Are there geographic variations in the rate of COVID-19 spread?
Is there any evidence to suggest geographic based virus mutations? |
| Task 6:
Relevant factors | Are inter/inner travel restrictions effective?
Are multifactorial strategies effective to prevent secondary transmission?
How does temperature and humidity affect the transmission of 2019-nCoV?
Is case isolation effective?
Is community contact reduction effective?
Is personal protective equipment effective?
Is school distancing effective?
Is the transmission seasonal?
Is workplace distancing effective?
Significant changes in transmissibility in changing seasons? |
| Task 7:
Models and open
questions | Are there changes in COVID-19 as the virus evolves?
Are there studies about phenotypic change?
Are there studies to monitor potential adaptations?
What do models for transmission predict?
What is known about mutations of the virus?
What is the human immune response to COVID-19?
What regional genetic variations (mutations) exist? |

Table 2: Questions submitted to RECORD for the tasks going from 3 to 7.

| Task | Question |
|------|----------|
| Task 8:
Patient descriptions | Can asymptomatic transmission occur during incubation?
How many pediatric patients were asymptomatic?
Is COVID-19 associated with cardiomyopathy and cardiac arrest?
What do we know about disease models?
What is the incubation period across different age groups?
What is the Incubation Period of the Virus?
What is the length of viral shedding after illness onset?
What is the longest duration of viral shedding?
What is the median viral shedding duration?
What is the natural history of the virus from an infected person?
Which is the proportion of patients who were asymptomatic? |
| Task 9:
Population studies | How to communicate with health care workers?
How to interact with high-risk elderly people?
What is the best management of patients who are underhoused or otherwise lower socioeconomic status?
Best modes of communicating with target high-risk populations?
What are recommendations for combating and overcoming resource failures?
What are ways to create hospital infrastructure to prevent nosocomial outbreaks and protect uninfected patients? |
| Task 10:
Material studies | How about adhesion to hydrophilic or phobic surfaces?
How about decontamination based on physical science?
How does the virus persist on different materials?
Is there susceptibility to environmental cleaning agents?
What do we know about viral shedding in blood?
What do we know about viral shedding in stool?
What do we know about viral shedding in urine?
What do we know about viral shedding nasopharynx? |
| Task 11:
Miscellaneous | Is there more than one strain in circulation?
Are there methods to control the spread in communities?
Which efforts have been made to identify the underlying drivers of fear?
Are there oral medications that might potentially work?
Which are the best ways of communicating with target high-risk populations? |

Table 3: Questions submitted to RECORD for the tasks going from 8 to 11.

| Task | Question |
|---|---|
| Questions formulated by *Centro Medico Sant'Agostino* | Is CT scan a reliable tool to detect the presence of COVID-19 infection? |
| | Which serological rapid test is shown to be the most reliable (in terms of specificity and sensitivity) to detect the presence of COVID-19 infection? |
| | Which is the average duration of the incubation period of COVID-19 virus? |
| | Does the duration of the incubation period of COVID-19 virus depend on individual characteristics (such as age, gender, comorbidities, etc.)? |
| | Does the viral load affect the severity of symptoms from COVID-19? |
| | Is there scientific evidence that flu vaccine prevents the infection from COVID-19? |
| | Is there scientific evidence that some blood types are more prone to be infected by COVID-19? |
| | Are tracking apps an effective tool to prevent the spread of COVID-19? |
| | Is there scientific evidence that warm weather reduces the spread of COVID-19? |
| | Is there scientific evidence that conjunctivitis is a symptom of COVID-19? |
| | What is the false positive and false negative rate in Diasorin serological rapid test? |
| | Are the antibodies IgM an effective measure to detect the presence of COVID-19? |
| | What is the average persistence of IgM antibodies in the blood, for individuals infected by COVID-19? |
| | Is there scientific evidence that some ethnic groups are more affected by COVID-19? |
| | Which comorbidities are responsible for more severe clinical conditions caused by COVID-19? |
| | How many days, on average, does the intensive care treatment last, for individuals infected by COVID-19? |

Table 4: Questions submitted to RECORD formulated and scored by *Centro Medico Sant'Agostino*.