# OpenReview forum: "RECORD: a Question Answering tool for COVID-19"
_EMNLP/2020/Workshop/NLP-COVID — Submitted to NLP-COVID19-EMNLP_

### Official Review · AnonReviewer1 · 2020-09-14
**Work shows potential, but not detailed enough to be accepted yet - needs more work.**

**Rating:** 4
**Confidence:** 4

**Review:**

This paper builds a question-answering model which answers questions from the recently available CORD-19 dataset. While the work promised potential, major steps are missing in both experimentation and writing.

Comments/ Suggested improvements:

1. In the pre-processing step of the paper the authors mention they use a list of keywords identifying the COVID-19 virus. However, this list of keywords needs to be released or revealed to facilitate repeatability and better understanding of the pre-processing step.

2. Similarly - the authors mention using k number of tokens less then 128 owing to the maximum capacity of the Sentence-BERT network. However, there is confusion in the above step. The authors mention that they choose a k-value slightly less than 128, however, in the same paragraph they also mention that they choose a k-value sufficiently smaller than 128. For repeatability purposes and for better understanding, revealing this number is required.

3. The literature review section of the paper lacks specific numbers. This is a rather crucial detail which is lacking. Since the primary contribution of the paper is an artificially intelligent question answering model which works on existing datasets, it is important to know exactly where this model stands in regards to a performance benchmark against current networks. It has been mentioned by the authors that owing to a lack of time and resources an empirical evaluation has been provided instead of a quantitative one, it does not show how the model in the paper stands up to existing work.

4. Section 2.3 mentions RECORD (the model proposed) extracts additional metadata such as title, authors, and so on to highlight reliability. Is this so that the person asking the questions knows that the answers are from a published source? Shouldn't that information already be clear owing to the dataset the model was trained upon? If that is not the case, then how does the additional data provide reliability? This needs to be specified in detail.

5. In Section 4.1 - the authors mention that humans were asked to annotate a score for each provided answer. How were the annotators selected? Were they annotators familiar with the model or the dataset or current literature and a ground truth? Was there any inter-annotator agreement between scores on a particular answer? If so, what was that score? Or did one annotator score only one answer?

6. While the questions are detailed, it is suggested that the authors provide some of the sample answers from each of the classes. Maybe a table showing a few questions with answers spanning scores 0-3 would be a very useful addition to this work.


Pros of the paper:

1. All the questions and all tasks are very well organized and detailed in appendix A.
2. The bar charts and graphs provided are clear to understand and shows the claimed performance benchmark.
3. The work is topical and promising. There is a lot of scope to improve and contribute to future venues.


In conclusion, although the work described in the paper promises to be full of potential, the paper lacks a lot of crucial details. The paper has very well presented supplementary material. All questions and tasks they are listed under are extremely important. The performance of 61.3% needs to be evaluated against other pre-existing benchmarks.

---

### Official Review · AnonReviewer2 · 2020-09-17
**Lacking in experimental details**

**Rating:** 4
**Confidence:** 4

**Review:**

This paper describes a QA tool for COVID-19 built on top of two components, a Sentence-BERT model for filtering papers by relevance, and a BERT-based QA model tuned on SQuAD. The authors test their model on two sets of questions, one created by the authors, and another set contributed by a regional healthcare facility, asking for human evaluators to score model results on a 4-point scale. The list of questions provided in Tables 1-4 are a good contribution, though the paper is lacking in implementation and experimental details.

More detailed feedback below:
1. The paper lacks experimental results, though the authors repeatedly refer to empirical comparisons. For the filtering step, the authors use Sentence-BERT on chunked papers, but do not compare to other baseline retrieval models, such as Anserini, CovidBERT, BioBERT, SciBERT etc, all of which the authors mention. The authors say Sentence-BERT was empirically better, but there are no results to support this. How much better? Similarly for QA, there are no numerical results showing that one model is better than another, just a statement that comparisons were performed. All modeling decisions therefore seem incredibly arbitrary.
2. There is also no way to understand how well the model performed. (Again, no baselines). The authors report evaluation results, but are these numbers reasonable? Are the methods reported here comparable to what would be expected in a general domain setting for example? Or the COVID-19 domain? There are datasets on COVID QA (Tang et al 2020) that could be evaluated on to give a sense of the performance of this model.

---

### Official Review · AnonReviewer3 · 2020-09-24
**Methodological weaknesses, needs more work**

**Rating:** 4
**Confidence:** 4

**Review:**

The paper presents a QA engine that takes COVID-19 related questions and outputs a span-based answer in a piece of text belonging to the relevant article. Article selection is based on finding out which textual chunks from the article is maximally similar to the question (using Sentence-BERT). When a set of articles relevant to the question is obtained, a general-domain BERT QA module outputs the span of text corresponding to the answer. The authors test these components as they are, without re-training or adaptation.

**Strengths of the paper:**
The pipeline separating retrieval and answer selection in general makes a lot of sense, and is clearly presented in the paper. Also, the authors consider a variety of COVID-19-related questions (from two different sources), and document them well in the appendix.

**Weaknesses:**
I see a number of methodological shortcomings that in my opinion make the publication of the article in the current state premature. I discuss those in the order as encountered while reading the paper.

I would be curious to see some statistics about how often the model selects the answer from a chunk (QA step) that was not previously selected (retrieval step) as the maximum scoring one.

The similarity calculation gives preference to locally maximally similar snippets of text with respect to the query. However, the importance of that snippet in the article has no weight by itself, so no indication of how central it is to that article. As an alternative, instead of just looking at the snippet maximising the cosine similarity, looking at the distribution of similarities could work better, e.g. when there are multiple snippets answering the query.
The authors mention that they carried out an empirical evaluation to single out DistilBERT as the best architecture, but do not mention how this evaluation (albeit “only” qualitative) was carried out. The same criticism applies to selecting the QA model. Because of this, there is little to learn from the paper---the comparison of model components was done only at face value.

Similarly, the answer evaluation has methodological issues, e.g. no mention of how many annotators graded the quality of answers, what was their agreement, how many answers were verified, and when the topic is deemed equivalent between the answer text and the question. Some examples could be provided for each answer score.

The authors go on to claim that “a substantial improvement to RECORD’s performances is brought by the analysis of the whole body of the articles”. But I’m wondering—improvement over what, and how was it measured? If the proposed tool can provide at least one precise answer ~60% of time, is this good or bad, and why? Further, more analysis could be done to show from which chunks (location in the text) the answers come from.

Finally, there is a lot of work involving QA for COVID-19, but the paper fails to address any of this work. See this as a starting point: https://openreview.net/pdf?id=0gLzHrE_t3z.